# The Perme Mobility Index: A new concept to assess mobility level in patients with coronavirus (COVID-19) infection

**Karina Tavares Timenetsky**[1], **Ary Serpa Neto**[1,2,3], **Ana Carolina Lazarin**[1], **Andreia Pardini**[1], **Carla Regina Sousa Moreira**[1], **Thiago Domingos Corrêa**[1], **Raquel Afonso Caserta Eid**[1], **Ricardo Kenji Nawa**[1] *

**1** Department of Critical Care Medicine, Hospital Israelita Albert Einstein, São Paulo, São Paulo, Brazil, **2** Australian and New Zealand Intensive Care-Research Centre (ANZIC-RC), Monash University, Melbourne, Australia, **3** Data Analytics Research & Evaluation (DARE) Centre, Austin Hospital and University of Melbourne, Melbourne, Victoria, Australia

* ricardo.nawa@einstein.br

**Data Availability Statement:** All relevant data are within the paper and its Supporting Information files.

## Abstract

### Introduction

The Coronavirus Disease 2019 (COVID-19) outbreak is evolving rapidly worldwide. Data on the mobility level of patients with COVID-19 in the intensive care unit (ICU) are needed.

### Objective

To describe the mobility level of patients with COVID-19 admitted to the ICU and to address factors associated with mobility level at the time of ICU discharge.

### Methods

Single center, retrospective cohort study. Consecutive patients admitted to the ICU with confirmed COVID-19 infection were analyzed. The mobility status was assessed by the Perme Score at admission and discharge from ICU with higher scores indicating higher mobility level. The Perme Mobility Index (PMI) was calculated [PMI = ΔPerme Score (*ICU discharge–ICU admission*)/ICU length of stay]. Based on the PMI, patients were divided into two groups: "Improved" (PMI > 0) and "Not improved" (PMI ≤ 0).

### Results

A total of 136 patients were included in this analysis. The hospital mortality rate was 16.2%. The Perme Score improved significantly when comparing ICU discharge with ICU admission [20.0 (7–28) points versus 7.0 (0–16) points; *P* < 0.001]. A total of 88 patients (64.7%) improved their mobility level during ICU stay, and the median PMI of these patients was 1.5 (0.6–3.4). Patients in the improved group had a lower duration of mechanical ventilation [10 (5–14) days versus 15 (8–24) days; *P* = 0.021], lower hospital length of stay [25 (12–37) days versus 30 (11–48) days; *P* < 0.001], and lower ICU and hospital mortality rate.

**Funding:** The author(s) received no specific funding for this work.

**Competing interests:** The authors have declared that no competing interests exist.

Independent predictors for mobility level were lower age, lower Charlson Comorbidity Index, and not having received renal replacement therapy.

## Conclusion

Patients' mobility level was low at ICU admission; however, most patients improved their mobility level during ICU stay. Risk factors associated with the mobility level were age, comorbidities, and use of renal replacement therapy.

## Introduction

Highly infectious and pathogenic, the outbreak coronavirus–severe acute respiratory syndrome coronavirus 2 (SARS-CoV-2), defined as the causal agent of Coronavirus Disease (COVID-19), was first identified and reported in Wuhan, Hubei province–China, in the late 2019 [1, 2]. The World Health Organization (WHO) declared COVID-19 as a pandemic in March 2020, raising an alert for an unprecedented public health emergency of international concern [2]. By mid-February 2021, more than 109 million cases had been confirmed in 223 countries [2].

Although COVID-19 mostly affects the respiratory system, these patients can present a wide spectrum of symptoms. In a large cohort of symptomatic patients with COVID-19 described early in the pandemic, 81% had mild disease, 14% had severe disease, and 5% became critically ill with organ failure; mortality in the critically ill group was 49% [3]. In the most critical cases, most patients evolve with respiratory failure, septic shock, and/or multiple organ dysfunction [3, 4]. Due to worsening of the respiratory function and oxygenation, 30 to 88% of critically ill patients require invasive ventilatory support, with a median time of mechanical ventilation (MV) ranging from 9 to 18 days [5–8].

As a result of prolonged use of invasive ventilatory support, use of deep sedation and neuromuscular blockade, COVID-19 patients submitted to invasive mechanical ventilation may remain restricted to bed for long periods, deprived of their daily activities. The inactivity and the hypercatabolic state observed in critically ill patients are associated with pronounced muscle mass loss, exceeding 10% during the first 7 days of admission to the intensive care unit (ICU) [9, 10]. Based on severe infections leading to respiratory distress in previous epidemics such as Severe Acute Respiratory Syndrome (SARS) and Middle East Respiratory Syndrome (MERS), whose patients presented physical function impairments for at least a year post ICU [11], patients with COVID-19 may have potential long-term secondary effects on musculoskeletal system, not yet described [12].

Several measuring instruments have been adapted to assess the physical function of patients admitted to the ICU [13]. Currently a total of six measures have been specifically developed for ICU settings [13]: Chelsea Critical Care Physical Assessment Tool (CPAx) [14], Physical Function Intensive Care Unit Test Scored (PFIT-s) [15], the Perme Intensive Care Unit Mobility Score (Perme Score) [16], Intensive Care Unit Mobility Scale (IMS) [17], Intensive Care Unit Optimal Mobility Score (SOMS) [18], and the Functional Status Score for the Intensive Care Unit (FSS-ICU) [19]. All these assessment measures evaluate patients' physical function; however, so far, only the Perme Score evaluates the barriers to mobilization. The study published by Wilches Luna et al. (2021) [20] established the minimal detectable change (MDC) of 1.36 points for the Perme Score, showing evidence of being sensitive to changes on patients' mobility level [20]. Previous studies have calculated the minimal detectable change of some physical

function instruments with the distribution-based method; the MDC was calculated for PFIT-s of 1.5 points [15], the CPAx established the value of 6.04 points [21], and the FSS-ICU the MDC of 2.0–5.0 points [22]. This determines the clinical significance of the results that are important to clinicians and researchers [20]. However, these findings do not consider the variation of the score during ICU length of stay (LOS) specifically; they only consider how many points of variation are needed to detect change in patients' mobility status. Therefore, this study will evaluate change in mobility level also considering the ICU LOS, thus testing a new concept entitled "Perme Mobility Index".

Measuring mobility early and longitudinally in the ICU is important to identify patients at risk of poor physical outcomes, monitor intervention efficacy, and to inform recovery trajectories [23–25].

The present study aims to describe the mobility level of patients admitted to the ICU diagnosed with COVID-19 infection, through a new concept entitled "Perme mobility index". As a secondary objective, we aimed to assess the factors associated with the mobility level at the time of ICU discharge.

## Materials and methods

### Study design and participants

This single-center retrospective cohort study was conducted in a tertiary private hospital located in the city of São Paulo, Brazil. The study was approved by the Institutional Review Board (IRB) of Hospital Israelita Albert Einstein's ethics committee under number CAAE: 30797520.6.0000.0071, and informed consent was waived. This study is reported in accordance with the Strengthening the Reporting of Observational studies in Epidemiology (STROBE) statement [26].

The clinical records of the first 200 consecutive patients admitted to the ICU with diagnosis of COVID-19 confirmed by reverse transcription–polymerase chain reaction (RT-PCR) for SARS-CoV-2 were considered eligible for the study. Key eligibility criteria included the following: 1) admission to the ICU, and 2) equal or older than 18 years. Study exclusion criteria consisted of patients who didn't present a report of mobility status on their electronic medical record at ICU admission and/or discharge.

### Data collection and study variables

All study data were retrieved from the electronic medical record (EMR) and from Epimed Monitor System® (Epimed Solutions, Rio de Janeiro, Brazil) hosted at Hospital Israelita Albert Einstein's servers, which is an electronic structured case report form where patients' data are prospectively entered by trained ICU case managers [27]. The ERM were accessed between March 1, 2020 to July 15, 2020. All data were extracted by an independent research assistant that did not participated in this study and were fully anonymized before been available for the researchers.

Collected variables included demographics, comorbidities, Simplified Acute Physiology Score (SAPS III score) at ICU admission–scores range from 0 to 217, with higher scores indicating more severe illness and higher risk of death [28], Sequential Organ Failure Assessment score (SOFA score) at ICU admission–scores range from 0 to 4 for each organ system, with higher aggregate scores indicating more severe organ dysfunction [29], Charlson Comorbidity Index–range from 0 to 5 for each comorbidity, with score of zero indicating that no comorbidities were found. The higher the score, the more likely the predicted outcome will result in mortality or higher resource use [30], Modified Frailty Index–categorized frailty using MFI values into non-frail (MFI = 0), pre-frail (MFI = 1–2) or frail (MFI ≥ 3) [31], use of invasive

mechanical ventilation, renal replacement therapy (RRT), and Extracorporeal Membrane Oxygenation (ECMO) at ICU admission and during ICU stay, need for tracheostomy, duration of mechanical ventilation, ICU and hospital length of stay, and ICU hospital mortality.

## Mobility status assessment

All consecutive patients admitted to the ICU with a confirmed diagnosis of COVID-19, who were assessed by a physical therapist, had their mobility status evaluated by the Perme Intensive Care Unit Mobility Score (Perme Score) [16] at two different moments: 1) ICU admission–within 24 hours of admission and 2) at ICU discharge, respectively. The Perme Score was proposed in order to assess the mobility status of patients admitted to intensive care [16]. It consists of 7 domains, as follows: 1) mental status, 2) potential mobility barriers, 3) functional strength, 4) bed mobility, 5) transfers, 6) gait, and 7) endurance. Starting with the assessment of patient's alertness observed upon arrival and initial contact with the rater, and ending with the total distance walked over a 2-minute period, the total score ranges from 0 to 32 points, with higher scores indicating higher mobility.

The Perme Mobility Index (PMI) is a new concept proposed in this study. It consists in calculating the difference between the total Perme Score at ICU discharge and the total Perme Score at ICU admission, divided by the ICU length of stay (ICU LOS) [PMI = ΔPerme Score (*ICU discharge–ICU admission*) / ICU LOS]. It is important to note that the PMI value can be either positive or negative. Positive values are associated with patients that improve their mobility status during ICU stay, whereas negative values are associated with patients that decrease their mobility status during ICU stay.

All COVID-19 patients admitted to the ICU are assessed by the physiotherapy team for an initial evaluation. The present institution has an early mobility protocol and patients are seen every day by a physical therapist. The Perme Score is part of the daily mobility status evaluation in the early mobility protocol. Due to the need for isolation in COVID-19 patients, therapies were performed only around the ICU beds. Therefore, all ICU beds are individually isolated, with enough space to perform out of bed exercises (around 82 square feet) while maintaining isolation during therapy.

## Statistical analysis

A convenience sample of the first 200 consecutive patients admitted to the ICU with a confirmed diagnosis of COVID-19 was considered for this analysis. Continuous variables are presented as median and interquartile range (IQR) values, and categorical variables as absolute and relative frequencies. Based on the PMI, patients were divided into two groups: "Improved" (PMI > 0) and "Not improved" (PMI ≤ 0) group. The Perme Score at ICU admission and discharge were compared between the groups using the Wilcoxon signed rank test.

Baseline and clinical characteristics of patients were compared between the groups using Fisher exact tests and Wilcoxon rank-sum tests. ICU and hospital length of stay were compared among the groups using sub-distribution hazard ratios derived from a Fine-Gray competing risk model, with death before the event being treated as competing risk [32]. ICU and hospital mortality were compared between the groups and reported as odds ratio (OR) and 95% confidence interval (CI) from generalized linear model with binomial distribution. The duration of ventilation was assessed as the median difference using quantile regression with T = 0.50, and results were estimated using bootstrap with 1,000 resamples.

A multivariable logistic regression model was used to identify factors independently associated with improvement in mobility. A list of candidate baseline predictors was determined *a priori* and included only variables with a known or suspected relationship with outcome. The

multivariable model was constructed considering variables with a $P < 0.05$ in the univariable analysis; it was confirmed using a backwards elimination technique and concluded with a final assessment for clinical and biological plausibility. Continuous missing predictors were present in less than 3% of the patients; thus, these values were imputed by median. Multicollinearity in the final models was assessed using variance-inflation factors, and linearity assumption of continuous variables was assessed using Box-Tidwell transformation considering the full model, testing the log-odds and the predictor variable. All analyses were conducted in R Version 3.6.3 (R Foundation) [33] and significance level was set at 0.05.

## Results

### Participants

From March 1 to May 31, 2020, 200 patients with confirmed COVID-19 were admitted to the ICU, of which a total of 136 (68%) met the inclusion criteria and were included in the present study. Sixty-four (32%) patients were excluded from the study due to missing data in one of the PMI (admission or discharge value). From 136 patients studied, 88 (64.7%) were included in the group "Improved" and 48 (35.3%) were included in the group "Not improved" (Table 1). Baseline characteristics of pooled patients are shown in Table 1.

Median (IQR) age of pooled patients was 69 (53–82) years; 57.4% were male, with a median (IQR) SAPS III of 53 (45–60); 47.8% were admitted to the ICU from the emergency department, and 60.3% received invasive mechanical ventilation (Table 1). Patients who improved their mobility level during the ICU stay were younger, had lower SAPS III score, lower Charlson comorbidity index, were less frail, and were more often admitted from the ward or transferred from other hospitals when compared to the patients who did not improve mobility (Table 1).

### Mobility

The Perme Score at ICU admission was similar between the two groups (Table 1). A total of 88 patients (64.7%) improved their mobility level during ICU stay (Table 1). The median (IQR) Perme Score total points in the pooled population study increased from ICU admission [7.0 (0.0–16) points] to ICU discharge [20.0 (7–28) points], respectively, $P = < 0.001$ (Table 1 and Fig 1).

Analyzing the PMI of the "improved" group, 49/88 (55.7%) of patients presented the PMI from +0.1 to +1.0, and 39/88 (44.3%) of patients presented the PMI $\geq$ +1.1. Regarding the PMI of the "not improved" group, 42/48 (87.5%) of patients presented the PMI from 0.0 to -1.0, and 6/48 (12.5%) presented the PMI $\leq$ -1.1 (Fig 2).

### Clinical outcomes

Patients' clinical outcomes are presented in Table 2. In the overall population, the median (IQR) duration of mechanical ventilation was 11 (7–18) days. Median duration of mechanical ventilation was lower in patients that improved the PMI compared to the group of patients that did not improve the PMI, [10 (5–14) days versus 15 (8–24) days], respectively, median difference, -5.39 (95% CI, -9.98 to -0.80); $P = 0.021$ (Table 2). The ICU and hospital length of stay were lower in patients that improved the PMI compared to the group of patients that did not improve the PMI, [11.5 (6.8–20.2) days versus 13.5 (8–26) days], median difference, 2.34 (95% CI, 1.5–3.66); $P = 0.001$, and [18 (13.5–32.5) days versus 25 (12–37) days], median difference, 4.77 (2.95–7.73); $P = 0.001$, respectively (Table 2). The ICU and hospital mortality rate were 16.2% and all deaths occurred in the group that did not improve mobility level (Table 2).

**Table 1. Baseline characteristics of the included patients.**

| Characteristic | Overall (*n* = 136) | Perme Mobility Index | | P Value |
|---|---|---|---|---|
| | | Improved (*n* = 88) | Not Improved (*n* = 48) | |
| Age (years) | 69 (53–82) | 62.5 (48.5–77) | 79.5 (62.5–86.2) | < 0.001 |
| Male gender–no. (%) | 78 (57.4) | 49 (55.7) | 29 (60.4) | 0.717 |
| Body mass index* | 28.3 (24.9–32.0) | 28.7 (24.9–32.5) | 27.8 (24.8–29.8) | 0.112 |
| Severity of illness | | | | |
| SAPS III score† | 53 (45–60) | 51.5 (44.8–57.2) | 58 (50.8–63) | 0.001 |
| SOFA score‡ | 4 (2–7) | 4 (1–6) | 4.5 (2–8) | 0.097 |
| Charlson comorbidity index§ | 1 (0–2) | 1 (0–1.2) | 2 (1–4) | 0.001 |
| Modified frailty index | 1 (0–3) | 1 (0–2) | 2 (1–3) | < 0.001 |
| Score | 0.1 (0–0.2) | 0.1 (0–0.2) | 0.2 (0.1–0.3) | < 0.001 |
| Clinical frailty–no. (%) | 12 (8.8) | 6 (6.8) | 6 (12.5) | 0.344 |
| ICU source of admission–no. (%) | | | | 0.034 |
| Emergency department | 65 (47.8) | 34 (38.6) | 31 (64.6) | |
| Ward | 54 (39.7) | 40 (45.5) | 14 (29.2) | |
| Step down unit | 5 (3.7) | 3 (3.4) | 2 (4.2) | |
| Other‖ | 12 (8.8) | 11 (12.5) | 1 (2.1) | |
| Organ support¶ –no. (%) | | | | |
| Non-invasive ventilation | 105 (77.2) | 71 (80.7) | 34 (70.8) | 0.205 |
| Invasive ventilation | 82 (60.3) | 53 (60.2) | 29 (60.4) | 0.999 |
| Endotracheal tube | 71 (86.6) | 49 (92.5) | 22 (75.9) | 0.046 |
| Tracheostomy | 11 (13.4) | 4 (7.5) | 7 (24.1) | |
| Renal replacement therapy | 31 (22.8) | 14 (15.9) | 17 (35.4) | 0.018 |
| ECMO | 0 (0.0) | 0 (0.0) | 0 (0.0) | - |
| Perme ICU Mobility Score** | | | | |
| At ICU admission | 7 (0.0–16) | 7 (0.0–15) | 8 (0.0–24) | 0.234 |
| At ICU final follow-up†† | 20 (7.0–28) | 24.5 (16–30) | 2 (0.0–11) | < 0.001 |
| Difference | 4.5 (0.0–16.2) | 13 (6.5–21) | 0.0 (-3.5–0.0) | < 0.001 |
| Perme mobility index | 0.6 (0.0–2.0) | 1.5 (0.6–3.4) | 0.0 (-0.5–0.0) | < 0.001 |
| Improved the PMI–no. (%) | 88 (64.7) | - | - | - |

*Definition of abbreviations*: SAPS: simplified acute physiology score; SOFA = sequential organ failure assessment; ICU = intensive care unit; ECMO = extracorporeal membrane oxygenation; PMI = Perme Mobility Index.

Data are median and interquartile range (IQR) values or n (%). Percentages may not total 100 because of rounding.

*The body-mass index (BMI) is calculated by weight in kilograms divided by the square of the height in meters.

†Scores on SAPS III range from 0 to 217, with higher scores indicating more severe illness and higher risk of death.

‡SOFA scores range from 0 to 4 for each organ system, with higher aggregate scores indicating more severe organ dysfunction.

§Charlson comorbidity index range from 0 to 5 for each comorbidity, with score of zero indicating that no comorbidities were found. The higher the score, the more likely the predicted outcome will result in mortality or higher resource use.

‖Other–includes other hospitals, ambulatory, procedure rooms, CT scan room, and other hospital units.

¶Organ support during ICU stay.

**Perme ICU mobility score range from 0 to 32, with higher scores indicating better mobility level.

††At ICU discharge or death.

## Factors associated with the mobility level

The univariable logistic regression analysis identified a total of five factors potentially associated with changes in the PMI (Table 3). In the multivariable logistic regression analysis, independent predictors for mobility level were the following: 1) lower age; 2) lower Charlson

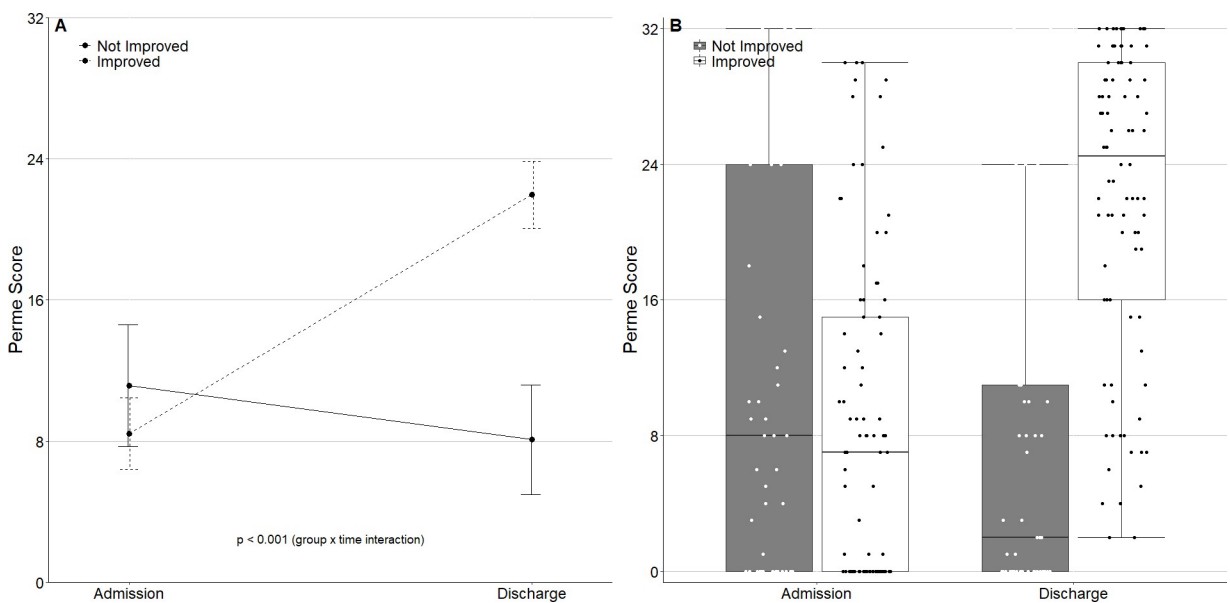

**Fig 1.** A. Mean and standard deviation of the Perme Score at ICU admission and discharge between patients that improved and did not improve the PMI during ICU stay (*light grey line* represents the group of patients that "not improved" and *dark grey line* represents the group of patients that "improved"). B. Boxplot of Perme Score between groups and between ICU admission and discharge in these groups (*light grey box* represents the group of patients that "not improved" and *dark grey box* represents the group of patients that "improved"). *Definition of abbreviations*: ICU = intensive care unit; PMI = perme mobility index; ICU LOS = intensive care unit length of stay.

Comorbidity Index; 3) not having received renal replacement therapy (Table 3). No multicollinearity was observed in the final model, data presented in S1 Table. Baseline characteristics of patients with or without missing in Perme Score are also presented in S2 and S3 Tables.

## Discussion

The main finding of this single-center cohort of critically ill COVID-19 patients was that approximately two thirds of patients submitted to invasive mechanical ventilation improved their mobility status during the ICU stay. We also observed that the main factors associated with decreased mobility level were increased age, number of comorbidities, and the need of renal replacement therapy.

The COVID-19 can be considered a multi-organ disease with potentially severe complications and musculoskeletal consequences, such as loss of muscle mass, muscle weakness, myopathies, and functional disabilities [11, 12]. To identify potential functional impairments, assessment of muscle mass, muscle strength, and physical function, the use of measurement instruments is recommended [13]. Consecutive assessments can provide objective, standardized, and comparable information when performed in distinct moments during hospital length of stay.

The Perme Score is an instrument specifically designed to evaluate patients' mobility status during ICU stay [16]. The PMI can be considered a new concept to better understand and interpret the mobility status of patients admitted to the ICU. The dimensionless reference PMI value of 1.0 means delta Perme is equal to the ICU LOS and can be used as a starting point for interpretation of clinical use of the PMI. Values above 1.0 determine a greater change of Perme score in a shorter period of time and should be interpreted as a robust increase of the mobility status during patients' ICU stay. On the other hand, values below 1.0 determine a

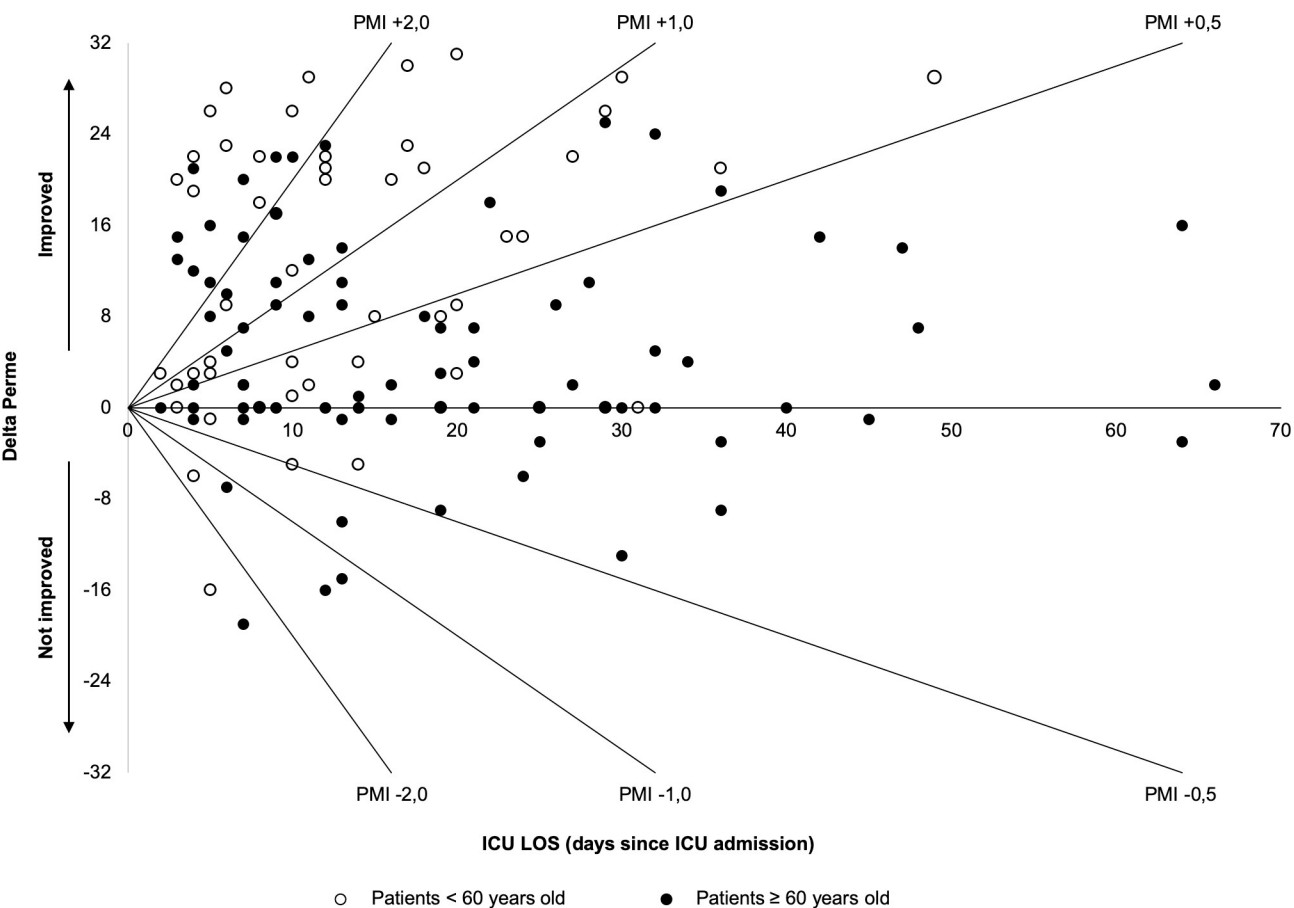

**Fig 2. Plot dispersion of Perme Mobility Index (PMI)–ΔPerme Score and the ICU LOS of each included patient (open circle represents patients under 60 years of age and closed circle represents patients above or equal 60 years of age).** The lines represent some of the PMI values, where it is possible to see that younger patients have higher PMI compared (concentrated at the top left of the graph) to older patients (concentrated at the bottom right of the graph), representing a better improvement of mobility level during the ICU LOS in younger patients. *Definition of abbreviations*: PMI = perme mobility index; ICU LOS = intensive care unit length of stay.

small change of Perme score during patients' ICU stay. The higher the value of PMI, the greater the increase in the mobility level during patients' ICU LOS.

It is important to note that changes measured using instruments over time can be confused with the clinimetric property entitled responsiveness–defined as the ability of an instrument to detect change over time [34]. There is also a minimum level of change, called the minimal detectable change (MDC), which can be identified by an instrument. It can be defined as the smallest alteration that the instrument is capable of detecting [35]. The study published by Wilches Luna et al. (2021) established the MDC of 1.36 points for the Perme Score, showing evidence of being sensitive to changes on patients' mobility level [20]. This determines the clinical significance of the results important to clinicians and researchers [20].

The main difference between the PMI and MCD is that the index analyzes the variation of Perme Score at a specific moment in time. The MDC determines how many points of variation are needed to detect change in the patient mobility status. It is highly recommended to always analyze the PMI values considering the ICU LOS in order to avoid misinterpretations of two different situations. The magnitude of the index must be analyzed individually, considering that the LOS can significantly affect the interpretation of the results. Literature references

**Table 2. Clinical outcomes of the included patients.**

| | | Perme Mobility Index | | | |
|---|---|---|---|---|---|
| | Overall (*n* = 136) | Improved (*n* = 88) | Not Improved (*n* = 48) | Effect Estimate (95% CI) | *P* Value |
| Duration of invasive mechanical ventilation (days) | 11 (7–18) | 10 (5–14) | 15 (8–24) | -5.39 (-9.98 to -0.80)* | 0.021 |
| In survivors (days) | 10 (6–15) | 10 (5–14) | 9.5 (8–15) | | |
| Extubation–no. (%) | 52 (86.7) | 44 (89.8) | 8 (72.7) | 3.30 (0.59 to 16.48)[†] | 0.148 |
| ICU length of stay (days) | 12 (7–23.2) | 11.5 (6.8–20.2) | 13.5 (8–26) | 2.34 (1.50 to 3.66)[‡] | < 0.001 |
| In survivors (days) | 11 (6.2–21) | 11.5 (6.8–20.2) | 8.5 (6.2–24) | | |
| Hospital length of stay (days) | 19.5 (12.2–35) | 18 (13.5–32.5) | 25 (12–37) | 4.77 (2.95 to 7.73)[‡] | < 0.001 |
| In survivors (days) | 19.5 (12–35.2) | 18 (13.5–32.5) | 30 (11–48) | | |
| ICU mortality–no. (%) | 22 (16.2) | 0 (0.0) | 22 (45.8) | — | < 0.001[§] |
| Hospital mortality–no. (%) | 22 (16.2) | 0 (0.0) | 22 (45.8) | — | < 0.001[§] |

*Definition of abbreviations*: ICU = intensive care unit; CI = confidence interval.

Data are median and interquartile range (IQR) values or *n* (%). Percentages may not total 100 because of rounding.

*Median difference from a quantile model considering a T = 0.50 and an asymmetric Laplace distribution. *P* Values were extracted after 1,000 bootstrap samplings.

[†]Odds ratio from a logistic regression.

[‡]Subdistribution hazard ratio from a Fine-Gray competing risk model with death before discharge as a competing risk.

[§]*P* value was calculated with the use of Fisher exact test.

show that prolonged ICU LOS is associated with less favorable outcomes, such as higher 1-year mortality [36].

Preliminary findings in COVID-19 patients identified that the higher the number of preexisting comorbidities, the higher the chances of developing a severe form of the disease [37]. Similarly, the findings of our present study identified age and comorbidities evaluated by the Charlson Comorbidity Index [30] as factors that influence patient's mobility status. Iaccarino et al. showed that increased age and comorbidities are the most important determinants to predict mortality in COVID-19 patients [38].

The results of our study showed negative effects on mobility level in patients undergoing RRT. The presence of dialysis catheters is frequently considered to be a barrier to mobilizing patients in the ICU [39]. However, studies have demonstrated the safety and feasibility of promoting rehabilitation and early mobilization in ICU with femoral dialysis catheter [40, 41]. Patients undergoing renal replacement therapy often experience loss of muscle mass secondary to progression of kidney disease, a decrease in protein synthesis, an increase in proteolysis, acidosis, inflammation, and the use of corticosteroids, which contributes to a catabolic state [42, 43].

Older age and comorbidities are also non-modifiable independent risk factors for developing ICU-acquired weakness (ICU-AW) [44]. Up to 66% of patients admitted to the ICU will be diagnosed with ICU-AW [45]. The physical function is one of the domains affected by post-intensive care syndrome (PICS). PICS can be defined as a new or worsening impairment of any of the physical, cognitive or mental health domains after a critical illness that persists beyond ICU stay [46, 47]. The pathophysiology of ICU-AW remains incompletely understood; however, some studies attribute this weakness to complex functional alterations within the central nervous system, the peripheral nerves, and the myofibers [44]. At least 25% of patients experience loss of independence and require assistance for activities of daily living one year after ICU admission [48]. Although there is an institutional protocol for early mobilization–which may for example explain why the use of MV is not a factor that may or may not influence PMI results–its implementation as part of clinical routine in critical care can be challenging when considering the multidimensional barriers found in the ICU [49]. This can take on

**Table 3. Univariable and multivariate logistic regression analysis addressing risk factors associated with patients' mobility level (n = 136 patients).**

| | Univariate Analysis | | | Multivariate Analysis | | |
|---|---|---|---|---|---|---|
| | OR | 95% CI | P Value | OR | 95% CI | P Value |
| Age | 0.95 | (0.93–0.98) | <0.001 | 0.35 | (0.17–0.68) | 0.003 |
| Body mass index* | 1.06 | (0.99–1.14) | 0.096 | - | - | - |
| SAPS III score† | 0.95 | (0.91–0.98) | 0.004 | 1.79 | (0.94–3.56) | 0.082 |
| SOFA score‡ | 0.91 | (0.81–1.02) | 0.097 | - | - | - |
| Charlson comorbidity index§ | 0.67 | (0.53–0.83) | 0.001 | 0.59 | (0.37–0.90) | 0.017 |
| Duration of ventilation | 0.98 | (0.95–1.01) | 0.200 | - | - | - |
| Male gender | 0.82 | (0.40–1.68) | 0.594 | - | - | - |
| Clinical frailty | 0.51 | (0.15–1.73) | 0.271 | - | - | - |
| ICU source of admission | | | | | | |
| Other‖ | Reference | | | Reference | | |
| Emergency room | 0.24 | (0.05–0.80) | 0.034 | 0.44 | (0.09–1.79) | 0.282 |
| Ward | 0.61 | (0.13–2.23) | 0.488 | 0.78 | (0.15–3.26) | 0.751 |
| Use of non-invasive ventilation | 1.72 | (0.75–3.90) | 0.193 | - | - | - |
| Use of invasive ventilation | 0.99 | (0.48–2.03) | 0.983 | - | - | - |
| Airway device | | | | | | |
| Not intubated | Reference | | | | | |
| Endotracheal tube | 1.21 | (0.57–2.57) | 0.621 | - | - | - |
| Tracheostomy | 0.31 | (0.07–1.16) | 0.089 | - | - | - |
| Use of renal replacement therapy | 0.34 | (0.15–0.78) | 0.011 | 0.29 | (0.10–0.77) | 0.015 |
| ICU length of stay | 0.98 | (0.96–1.01) | 0.209 | - | - | - |

*Definition of abbreviations*: OR = odds ration; CI = confidence interval; SAPS = simplified acute physiology score; SOFA = sequential organ failure assessment;
ICU = intensive care unit.

Variables with $p < 0.05$ were selected for the multivariable model.

*The body-mass index (BMI) is the weight in kilograms divided by the square of the height in meters.

†Scores on SAPS III range from 0 to 217, with higher scores indicating more severe illness and higher risk of death.

‡SOFA scores range from 0 to 4 for each organ system, with higher aggregate scores indicating more severe organ dysfunction.

§Charlson comorbidity index range from 0 to 5 for each comorbidity, with score of zero indicating that no comorbidities were found. The higher the score, the more likely the predicted outcome will result in mortality or higher resource use.

‖Other–includes other hospitals, ambulatory, procedure rooms, CT scan room and other hospital units.

new dimensions if we consider the new challenges COVID-19 pandemic has confronted us with, such as an increased work demand, heavy workload pressure, stress associated with the risk of infection, and mental pressure [50].

The findings of this study are consistent with the literature in regards to the factors that affect the mobility status of patients admitted to the ICU. Nevertheless, the study has limitations that should be acknowledged. First, there is evidence of the correlation between the use of neuromuscular blockade and corticosteroid drugs and ICU-AW, yet our study did not provide these data, which might have contributed to our findings. Another limitation is the absence of a severity score during hemodialysis, which could help us elucidate the difference between patients who used RRT and improved the PMI and those who did not improve the PMI.

Based on the study's result, it was observed that COVID-19 pandemic not only causes respiratory impairment, but also affects patient's mobility level. Although patients improve their mobility level, they do not achieve the highest Perme Score at ICU discharge, which demonstrates they still need post ICU rehabilitation. Another important aspect regarding the results

in our study involves the associated factors to mobility level, such as age, comorbidities, and use of renal replacement therapy. This information may help physiotherapists to identify patients at higher risk of no improvement in mobility level and to include mobilization therapies earlier during ICU stay in this specific population, as therapy for COVID-19 patients should not only include respiratory therapy but also mobility therapy as early as possible. During this pandemic, there has been an increase in ICU demand, which may preclude the practice of adequate mobility therapy as in usual care. Therefore, it is interesting to stratify patients with higher risk of no improvement of mobility level during ICU stay.

To our knowledge, this is the first study to bring up this new concept of the mobility index, arising from the clinical use of the Perme Score. The change of mobility status over time is the main purpose of the PMI. Future studies should test and present more details regarding the clinical use of this index and its interpretation in order to contribute to a better understanding of clinical outcomes of critically ill patients. Our preliminary data suggest that the PMI can be used in clinical practice to help the multidisciplinary team to better understand the profile of patients' mobility level. Although most patients improved the PMI, the long-term impact on patients' physical function still remains unanswered. There is a need for follow-up of post-ICU COVID-19 patients to offer tailored rehabilitation.

Our study has limitations. First, the timeframe established: it was limited to ICU stay only. The study might have had additional relevance if the patients had been followed post-ICU admission and post-discharge. Second, the focus on a specific population: this study focused on evaluating the mobility level of COVID-19 patients. However, in order to validate this new concept of the PMI, other populations must be studied. Information regarding the impact of mobility level during hospital stay as well as long-term follow-up should be investigated in future studies.

## Conclusion

In this retrospective single-center cohort study, the mobility level in critically ill COVID-19 patients was low at ICU admission; however, most patients improved their mobility level during ICU stay. Risk factors associated with mobility level were age, comorbidities, and use of renal replacement therapy.

## Supporting information

**S1 Table. Assessment of multicollinearity in the final multivariable model.** *Definition of abbreviations*: SAPS = simplified acute physiology score; ICU = intensive care unit; CI = confidence interval; GVIF = generalized variance-inflation factors; df = degrees of freedom. *Scores on SAPS III range from 0 to 217, with higher scores indicating more severe illness and higher risk of death. †Charlson comorbidity index range from 0 to 5 for each comorbidity, with score of zero indicating that no comorbidities were found. The higher the score, the more likely the predicted outcome will result in mortality or higher resource use. (DOCX)

**S2 Table. Baseline characteristics of patients with or without missing in Perme Score.** *Definition of abbreviations*: SAPS: simplified acute physiology score; SOFA = sequential organ failure assessment; ICU = intensive care unit; ECMO = extracorporeal membrane oxygenation; PMI = perme mobility index. Data are median and interquartile range (IQR) values or n (%). Percentages may not total 100 because of rounding. *The body-mass index (BMI) is calculated by weight in kilograms divided by the square of the height in meters. †Scores on SAPS III range from 0 to 217, with higher scores indicating more severe illness and higher risk of death.

[‡]SOFA scores range from 0 to 4 for each organ system, with higher aggregate scores indicating more severe organ dysfunction. [§]Charlson comorbidity index range from 0 to 5 for each comorbidity, with score of zero indicating that no comorbidities were found. The higher the score, the more likely the predicted outcome will result in mortality or higher resource use. [‖]Other–includes other hospitals, ambulatory, procedure rooms, CT scan room, and other hospital units. [¶]Organ support during ICU stay. [**]Perme ICU mobility score range from 0 to 32, with higher scores indicating better mobility level.[††]At ICU discharge or death.
(DOCX)

**S3 Table. Clinical outcomes in patients with or without missing in Perme Score.** *Definition of abbreviations*: ICU = intensive care unit. Data are median and interquartile range (IQR) values or *n* (%). Percentages may not total 100 because of rounding.
(DOCX)

**S1 Dataset.**
(XLSX)

## Acknowledgments

We thank the doctors, nursing staff, physical therapists, and all members of the multidisciplinary team of Hospital Israelita Albert Einstein who managed patients during the SARS-CoV-2 outbreak. The authors thank Helena Spalic for proofreading this manuscript.

## Author Contributions

**Conceptualization:** Karina Tavares Timenetsky, Ricardo Kenji Nawa.

**Data curation:** Ary Serpa Neto, Andreia Pardini.

**Formal analysis:** Ary Serpa Neto.

**Investigation:** Karina Tavares Timenetsky, Ricardo Kenji Nawa.

**Methodology:** Karina Tavares Timenetsky, Ary Serpa Neto, Ricardo Kenji Nawa.

**Project administration:** Karina Tavares Timenetsky, Ricardo Kenji Nawa.

**Supervision:** Karina Tavares Timenetsky, Ricardo Kenji Nawa.

**Validation:** Karina Tavares Timenetsky, Ary Serpa Neto, Ana Carolina Lazarin, Thiago Domingos Corrêa, Ricardo Kenji Nawa.

**Writing – original draft:** Karina Tavares Timenetsky, Ana Carolina Lazarin, Ricardo Kenji Nawa.

**Writing – review & editing:** Ary Serpa Neto, Carla Regina Sousa Moreira, Thiago Domingos Corrêa, Raquel Afonso Caserta Eid.

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
