## [Decision Letter · Decision Letter 0]

11 Feb 2021

PONE-D-21-01443

Analysis of the mobility level of patients admitted to the intensive care unit with coronavirus (COVID-19) infection

PLOS ONE

Dear Dr. Nawa,

Thank you for submitting your manuscript to PLOS ONE. After careful consideration, we feel that it has merit but does not fully meet PLOS ONE’s publication criteria as it currently stands. Therefore, we invite you to submit a revised version of the manuscript that addresses the points raised during the review process.

We look forward to receiving your revised manuscript.

Kind regards,

Aleksandar R. Zivkovic

Academic Editor

PLOS ONE

2. In the ethics statement in the manuscript and in the online submission form, please provide additional information about the patient records/samples used in your retrospective study, including: a) whether all data were fully anonymized before you accessed them; b) the date range (month and year) during which patients' medical records/samples were accessed; c) the source of the medical records/samples analyzed in this work (e.g. hospital, institution or medical center name).

3. Thank you for including your ethics statement:  "N/A".   

Reviewers' comments:

Reviewer's Responses to Questions

**Comments to the Author**

1. Is the manuscript technically sound, and do the data support the conclusions?

Reviewer #1: Yes

Reviewer #2: Yes

Reviewer #3: No

Reviewer #4: Yes

2. Has the statistical analysis been performed appropriately and rigorously? 

Reviewer #1: Yes

Reviewer #2: Yes

Reviewer #3: I Don't Know

Reviewer #4: Yes

3. Have the authors made all data underlying the findings in their manuscript fully available?

Reviewer #1: Yes

Reviewer #2: Yes

Reviewer #3: Yes

Reviewer #4: Yes

4. Is the manuscript presented in an intelligible fashion and written in standard English?

Reviewer #1: Yes

Reviewer #2: Yes

Reviewer #3: Yes

Reviewer #4: Yes

5. Review Comments to the Author

Reviewer #1: I appreciate the invitation to review the article. This is an interesting article, but it has some aspects that can improve. I respectfully attach my comments.

Authors might revise grammatic aspects of the manuscript.

Title

The title might include the index used to identify the mobility level. Avoid the use of many articles in the title.

Introduction

It clearly justifies the interest of the chosen topic in the studied population and highlights the importance of the study. The authors might think of emphasize in the introduction the new concept proposed regarding the Index.

Methods

The study design is a retrospective cohort study, given that the information was taken from electronic medical records, the authors should revise the wording of the inclusion and exclusion criteria to avoid confusion regarding the temporality of the study. For example, instead of “Patients admitted to the ICU with diagnosis of COVID-19…”, the authors might consider change it to “the clinical records of patients admitted to the ICU with a confirmed diagnosis of COVID 19…”

Authors must describe the process of construction/validation of the new index presented in this study “The Perme Mobility Index (PMI)”. This is a very important contribution and given that it has never been done, a validation process is necessary.

Discussion

Authors also, might consider using the minimum detectable difference of the Perme Score that already published in the discussion section.

Reviewer #2: General comments:

In this study, the authors use average change in Perme score (a measure of mobility in critical care) per day in ICU (called the Perme Mobility Index, PMI) to describe the change in mobility in patients admitted to the ICU with COVID-19. Independent predictors for the PMI were determined.

This report is relevant and well-presented. It is well-organized and the figures and tables present the data in a useful fashion. My biggest question is: what is the clinical significance of the topic and what are the clinical implications of the results? I feel that these aspects could be developed further, so that rather than just describing the patients, the results can be used to further patient care.

Please proof-read for grammar before resubmission.

Specific comments:

Abstract

What is the Perme score out of? Do higher numbers signify better or worse?

If you introduce an acronym, continue using it (e.g. RRT)

Introduction

“Thus far …” – when exactly was this date?

A greater justification of why it’s important to assess mobility in ICU patients and what the potential clinical relevance is would be good to introduce. For example, is there some way to mitigate the detrimental effect of decreased mobility in the ICU?

Material and methods – Study design and participants

Define RT-PCR

Under “Mobility status assessment” you mention “All patients admitted to the ICU with referral to physical therapy …”. Should having a referral to physical therapy be an inclusion criterion? What proportion of patients in the ICU with COVID-19 is referred for physical therapy?

How did you decide on the sample size?

Materials and methods – Data collection and study variables

Confirm when each variable was recorded. Some are mentioned at ICU admission, but others (ex. Charlson Comorbidity Index) do not say when they were recorded or if they changed over admission.

Add details about the collected variables, what are they out of, if high score is better or worse, what the score assesses.

Mention that you included individuals that died in the ICU (it is included later but would be useful to have here too).

Materials and methods - Mobility status assessment

The PMI is similar to FIM (Functional Independence Measure) Efficiency, which is the average change in FIM score per day of admission.

Results – participants

No comments.

Discussion

Instead of “Values above 1.0 determine a greater variation of delta …”, perhaps “… a greater change in Perme …” would be better. Same thing for values below 1.0.

What PMI has been seen in other ICU populations? Even if this metric is new, you can calculate it from the data provided in other studies.

“ICU-AW” is inconsistently written.

What is the significance of the overall results with respect to the COVID-19 pandemic? Will different variants have different effects on the change in mobility after ICU stay? Does this affect recommendations for in-ICU therapy or post-ICU therapy? See the general comments as well.

Limitations

How did the inclusion of patients who died influence the results?

One limitation is the limited time-frame of the study. It would have added more relevance if the patients had been followed post-ICU admission and post-discharge. You mention this in the last paragraph on page 14 but it should be mentioned as a limitation.

Reviewer #3: Dear authors

Thank you for this interesting study. I do have several comments.

Ethics Statement: you write N/A but the study involves data of human participants. Then in the paper (page 4) you write that the consensus was waved so I gues you did get ethical approval? Please clarify. I do not know Brazilian regulations but in many countries consensus would be required.

In the introduction I would appreciate more info on comparable scales or indexes ans the rationale why to develop the PMI.

Also because in case a scale already exists you need to compare. If it lacks then you ahve motive to develop yours.

On page 5 under "Mobility status assessment" you mention that you inluded patienst "with referral to physical therapy". Please expand: what are the indications or criteria to refer patients? Becaues this may creat a bias in your sample.

Information on the PT and/or larger Rehabilitation treatment of the patients is totally lacking. This would clearly influence the mobility of the patients and its evolution.

Also specific aspects regarding COVID-19 sucha as isolation, PPE use etc are very linked and you provide no information at all. This may be very different from other populations with infectious disease. Also the exercise tolreance of covid patients seems different from otehr ICU patients. Please mention that with some references.

Statistical analysis

Here I have main concerns. As the PERM score is an ordinal scale it probably needs RASCH analysis. Given the data re not continous you are not sure that one degree of improvement may vlaue another and that you would be comparing tthe same difference.

Also we need information on the clinically significatn difference to decide when a patient is considere Improved or not improved.

How did you deal with patients who died? Did you have the scores at Death? are they considered as score at discharge?

In the Results section, page 9 first paragraph I do not understand. You mean from 7 to 20?

Factors associatesd with mobility level: Data on therapy, Rehab treatment, PT etc are lacking.

Discussion

You need to compare with possibly existing other studies or other instruments on the same topic

I do not understand your considerations on LOS page 12 in the second paragraph.

Limitations: the fact that this study considers only covid patients which means a very specific context; would need to be validated also in other populations

Reviewer #4: This was a retrospective observational study of 136 patients admitted to an ICU in single-center in Brazil with COVID. The authors found an association between improved mobility scores and better outcomes. Predictors of better mobility scores were identified. The included patients were divided into two groups based on a novel scoring introduced in this study called the PMI. The results were presented by PMI group.

An exclusion criteria is patients without mobility status report at ICU admission and/or discharge

There is a potential for significant selection bias by excluding these patients. For example, patients who died during ICU stay may not have a mobility status at “ICU discharge”.

It seems that only patients who were referred to physical therapy during ICU stay had their mobility levels assessed using the Perme score and were included in this study. Patients who were not referred to physical therapy were excluded. This introduces a potential source of selection bias that must be addressed.

There were 64 (32%) patients excluded from the study due to missing data. Comparing the clinical outcomes in these patients (without mobility data) to those included in the study (with mobility data) would be helpful at addressing the above points. Was the mortality rate or length of stay similar? How about other key clinical datapoints?

The methods state that a “convenience sample” was used. If the study was done retrospectively with ethics approval, why were consecutive patients not included?

Page 10: This sentence seems to be missing a final word: “No multicollinearity was observed in the final model, data presented in (S1 Table).”

Page 14: Should be “purpose” in the sentence: “The change of mobility status over time is the main propose of the PMI.

6. PLOS authors have the option to publish the peer review history of their article (what does this mean?). If published, this will include your full peer review and any attached files.

Reviewer #1: No

Reviewer #2: No

Reviewer #3: **Yes: **Carlotte Kiekens

Reviewer #4: No

---

## [Author Response · Author response to Decision Letter 0]

30 Mar 2021

Reviewer #1:

I appreciate the invitation to review the article. This is an interesting article, but it has some aspects that can improve. I respectfully attach my comments.

Dear Reviewer #1,

Thank you for your hard work reviewing this manuscript. We appreciate the time spent revising the manuscript. Please find below the answers to all your comments and suggestions.

Authors might revise grammatic aspects of the manuscript.

Thank you for your careful revision identifying some grammatical aspects. We submitted the manuscript for the proofreader to review its full content.

Title

The title might include the index used to identify the mobility level. Avoid the use of many articles in the title.

Thank you for your comment. We appreciate your suggestion to include the index used to identify patients’ mobility level. Taking into account your suggestion, we have decided to rewrite the title as follows: “The Perme Mobility Index: a new concept to assess mobility level in patients with coronavirus (COVID-19) infection”.

Introduction

It clearly justifies the interest of the chosen topic in the studied population and highlights the importance of the study. The authors might think of emphasize in the introduction the new concept proposed regarding the Index.

Thank you for your suggestion. We revised the introduction section and decided to emphasize the new concept proposed regarding the “Perme Mobility Index – PMI”, rewording the sentence, as follows: “The present study aims to describe the mobility level of patients admitted to the ICU diagnosed with COVID-19 infection through a new concept entitled "Perme mobility index". As a secondary objective, we aimed to assess the factors associated with the mobility level at the time of ICU discharge.”

Methods

The study design is a retrospective cohort study, given that the information was taken from electronic medical records, the authors should revise the wording of the inclusion and exclusion criteria to avoid confusion regarding the temporality of the study. For example, instead of “Patients admitted to the ICU with diagnosis of COVID-19…”, the authors might consider change it to “the clinical records of patients admitted to the ICU with a confirmed diagnosis of COVID 19…”

Thank you for your suggestion. In order to avoid misunderstanding of the inclusion and exclusion criteria of the study, we have rewritten the sentence as follows: “The clinical records of the first 200 consecutive patients admitted to the ICU with diagnosis of COVID-19 confirmed by reverse transcription–polymerase chain reaction (RT-PCR) for SARS-CoV-2 were considered eligible for the study. Key eligibility criteria included the following: 1) admission to the ICU and 2) equal or older than 18 years. Study exclusion criteria consisted of patients who didn’t present a report of mobility status on their electronic medical record at ICU admission and/or discharge”.

Authors must describe the process of construction/validation of the new index presented in this study “The Perme Mobility Index (PMI)”. This is a very important contribution and given that it has never been done, a validation process is necessary.

Thank you for your suggestion. The “Perme mobility index” has not yet been fully explored and tested for its clinical use. In our opinion, because this is the first study to discuss this new concept arising from the clinical use of the Perme Score, future studies are still necessary to provide additional information on the following topics:

• Validate the PMI;

• Test the clinical use of PMI for populations other than COVID patients;

• Test the predictive validity of the PMI;

• Establish the stratification risk based on the index at different levels of mobility;

• Establish range values for clinical use (e.g.: + 0.0 <PMI < + 1.0; the patient present… + 1.0, PMI < + 2.0; the patient present…)

Discussion

Authors also, might consider using the minimum detectable difference of the Perme Score that already published in the discussion section.

Thank you for your comment and suggestion to include additional information on the minimum detectable difference (MDC) of the Perme Score. We decided to include the study published by Wilches Luna et al. (2021) in the discussion section of the manuscript as suggested.

See below the full paragraph after the revision process: “The Perme Score is an instrument specifically designed to evaluate patients’ mobility status during ICU stay [16]. The PMI can be considered a new concept to better understand and interpret the mobility status of patients admitted to the ICU. The dimensionless reference PMI value of 1.0 means delta Perme is equal to the ICU LOS and it can be used as a starting point for interpretation of clinical use of the PMI. Values above 1.0 determine a greater change of Perme score in a shorter period of time and should be interpreted as a robust increase of the mobility status during patients’ ICU stay. On the other hand, values below 1.0 determine a small change of Perme score during patients’ ICU stay. The higher the value of PMI, the greater the increase in the mobility level during patients’ ICU LOS.

It is important to note that changes measured using instruments over time can be confused with the clinimetric property entitled responsiveness – defined as the ability of an instrument to detect change over time [35]. There is also a minimum level of change, called the minimal detectable change (MDC), which can be defined by an instrument. It can be defined as the smallest alteration that the instrument is capable of detecting [36]. The study published by Wilches Luna et al. (2021) established the MDC of 1.36 points for the Perme Score, showing evidence of being sensitive to changes in patients’ mobility level [20]. This determines the clinical significance of the results important to clinicians and researchers [20].

The main difference between the PMI and MCD is that the index analyzes the variation of Perme Score at a specific moment in time. The MDC determines how many points of variation are needed to detect change in patients’ mobility status. It is highly recommended to always analyze the PMI values considering the ICU LOS in order to avoid misinterpretations of two different situations., Considering that the LOS can significantly affect the interpretation of the results, the magnitude of the index must be analyzed individually. Literature references show that prolonged ICU LOS is associated with less favorable outcomes, such as higher 1-year mortality [37].

Reviewer #2:

General comments: In this study, the authors use average change in Perme score (a measure of mobility in critical care) per day in ICU (called the Perme Mobility Index, PMI) to describe the change in mobility in patients admitted to the ICU with COVID-19. Independent predictors for the PMI were determined.

Dear Reviewer #2,

Thank you for the time spent reviewing this manuscript. Below, you will find the answers point-by-point for all your comments and suggestions.

This report is relevant and well-presented. It is well-organized and the figures and tables present the data in a useful fashion. My biggest question is: what is the clinical significance of the topic and what are the clinical implications of the results? I feel that these aspects could be developed further, so that rather than just describing the patients, the results can be used to further patient care.

Thank you for your comment.

The clinical implications of the results of the PMI are that they allow therapists to evaluate changes in patient’s mobility level over time, using the length of stay (LOS) in the index. The positive values of the PMI suggest improvement of patients’ mobility level over time, while the negative values or “zero” PMI result suggest non-improvement.

With this information, the therapist can evaluate the progress of patients’ mobility level during ICU stay and focus on the rehabilitation that is still needed during hospital stay in order to improve their mobility level before hospital discharge. Another clinical implication of this study is that during ICU stay, COVID-19 patients had respiratory impairment and mobility impairment, associated with age, comorbidities, and use of renal replacement therapy.

The associated factors found in our study may help physical therapists to identify patients at greater risk of experiencing no improvement in mobility level. They may also allow for an earlier inclusion of mobilization therapies during ICU stay in this specific population, as therapy for COVID-19 patients should not only include respiratory therapy but also mobility therapy as early as possible. During this pandemic, we have been facing a high demand in ICU, which may not allow for an adequate mobility therapy as in usual care. Therefore, it is interesting to stratify patients with greater risk of experiencing no improvement of mobility level during ICU stay.

Please proof-read for grammar before resubmission.

Thank you for your suggestion. We will have the entire manuscript proofread once again to avoid grammatical and typo errors.

Specific comments:

Abstract

What is the Perme score out of? Do higher numbers signify better or worse? If you introduce an acronym, continue using it (e.g. RRT)

Thank you for your comment and the opportunity to clarify this point.

A high total point of the Perme score indicates few potential mobility barriers and decreased assistance, whereas a low score indicates more potential barriers to mobility and more assistance needed for mobility (Perme C, Nawa RK, Winkelman C, Masud F. A tool to assess mobility status in critically ill patients: the Perme Intensive Care Unit Mobility Score. Methodist Debakey Cardiovasc J 2014; 10:41-9).

We have revised the abstract including additional information regarding the Perme score as suggested. Our revised sentence is as follows: “The mobility status was assessed by the Perme Score at admission and discharge from ICU with higher scores indicating higher mobility level. The Perme Mobility Index (PMI) was calculated [PMI = �Perme Score (ICU discharge – ICU admission)/ICU length of stay]. Based on the PMI, patients were divided into two groups: “Improved” (PMI > 0); or “Not improved” (PMI ≤ 0)”.

We also carefully revised the entire manuscript to introduce the acronyms and to continue using it consistently.

Introduction

“Thus far …” – when exactly was this date?

Thank you for your comment. We have updated this information in the manuscript as follows: “By mid-February 2021, more than 109 million cases had been confirmed in 223 countries [2].”

A greater justification of why it’s important to assess mobility in ICU patients and what the potential clinical relevance is would be good to introduce. For example, is there some way to mitigate the detrimental effect of decreased mobility in the ICU?

Thank you for your comment. The following sentence was included in the Introduction section to justify its importance: “Measuring mobility early and longitudinally in the ICU is important to identify patients at risk of poor physical outcomes, to monitor intervention efficacy, and to inform recovery trajectories [24-26].”

Material and methods – Study design and participants

Define RT-PCR

Thank you for your comment.

We spelled out the abbreviation “RT-PCR” as “Reverse transcription polymerase chain reaction”. Reverse transcription polymerase chain reaction (RT-PCR) is a variation of standard PCR that involves the amplification of specific mRNA obtained from small samples. It is how the COVID-19 diagnosis test has been conducted at Hospital Israelita Albert Einstein.

We have rewritten the sentence in the manuscript as follows: “Patients admitted to the ICU with diagnosis of COVID-19 confirmed by reverse transcription–polymerase chain reaction (RT-PCR) for SARS-CoV-2 were considered eligible for the study”.

Under “Mobility status assessment” you mention “All patients admitted to the ICU with referral to physical therapy …”. Should having a referral to physical therapy be an inclusion criterion?

Thank you for your comment. No. The referral to physical therapy was not an inclusion criterion. We apologize for the misinterpretation of the sentence. We decided to rewrite the sentence as follows: “All consecutive patients admitted to the ICU with a confirmed diagnosis of COVID-19 were assessed by a physical therapist, had their mobility status evaluated by the Perme Intensive Care Unit Mobility Score (Perme Score) (…)”

One major limitation of retrospective studies is missing data. This may prevent us from including 100% of the information needed for statistical analysis.

What proportion of patients in the ICU with COVID-19 is referred for physical therapy?

The proportion is 100%. All patients admitted to the ICU with COVID-19 are assessed by a physical therapist due to the need of respiratory and motor evaluation and treatment. In Brazil, we do not have “respiratory therapist” professionals, like in other countries in the world. The physical therapist is the professional responsible for managing patients’ respiratory conditions as well as providing rehabilitation during ICU and hospital length of stay.

How did you decide on the sample size?

The sample size was based on the first 200 consecutive cases of patients admitted to the ICU with confirmed diagnosis of COVID-19 at Hospital Israelita Albert Einstein. We realized that we did not mention the “first 200 consecutive cases” on the first version of the manuscript. Thus, we have rewritten the sentence as follows: “A convenience sample of the first 200 consecutive patients admitted to the ICU with a confirmed diagnosis of COVID-19 was considered for this analysis”.

Materials and methods – Data collection and study variables

Confirm when each variable was recorded. Some are mentioned at ICU admission, but others (ex. Charlson Comorbidity Index) do not say when they were recorded or if they changed over admission.

Thank you for your comment and the opportunity to clarify this point.

The variables utilized to calculate the SAPS and SOFA scores are collected within 24 hours of ICU admission. All variables are uploaded to Epimed Monitor System® (Epimed Solutions, Rio de Janeiro, Brazil) reference: “Zampieri FG, Soares M, Borges LP, Salluh JIF, Ranzani OT. The Epimed Monitor ICU Database®: a cloud-based national registry for adult intensive care unit patients in Brazil. Epimed Monitor ICU Database®: um registro nacional baseado na nuvem, para pacientes adultos internados em unidades de terapia intensiva do Brasil. Rev Bras Ter Intensiva 2017;29:418-426.”. The Epimed Monitor System ®, calculate the SAPS score for severity of illness and the risk of death and the SOFA score for severity of organ dysfunction. The Charlson Comorbidity Index is also calculated by Epimed Monitor System®, according to data imputed on the Epimed® system.

Add details about the collected variables, what are they out of, if high score is better or worse, what the score assesses.

Thank you for your comment and suggestion to provide additional information about the collected variables. We have revised the paragraph including more details about the variables as follows: “Collected variables included demographics, comorbidities, Simpliﬁed Acute Physiology Score (SAPS III score) at ICU admission – scores range from 0 to 217, with higher scores indicating more severe illness and higher risk of death [15], Sequential Organ Failure Assessment score (SOFA score) at ICU admission – scores range from 0 to 4 for each organ system, with higher aggregate scores indicating more severe organ dysfunction [16], Charlson Comorbidity Index – range from 0 to 5 for each comorbidity, with score of zero indicating that no comorbidities were found. The higher the score, the more likely the predicted outcome will result in mortality or higher resource use [17], Modified Frailty Index – categorized frailty by using MFI values into non-frail (MFI = 0), pre-frail (MFI = 1–2) or frail (MFI ≥ 3) [18,], use of invasive mechanical ventilation, renal replacement therapy (RRT), and Extracorporeal Membrane Oxygenation (ECMO) at ICU admission and during ICU stay, need for tracheostomy, duration of mechanical ventilation, ICU and hospital length of stay, and ICU hospital mortality”.

Mention that you included individuals that died in the ICU (it is included later but would be useful to have here too).

Thank you for your comment. We have included this information as suggested.

Materials and methods - Mobility status assessment

The PMI is similar to FIM (Functional Independence Measure) Efficiency, which is the average change in FIM score per day of admission.

Thank you for your comment.

Results – participants

No comments.

Thank you.

Discussion

Instead of “Values above 1.0 determine a greater variation of delta …”, perhaps “… a greater change in Perme …” would be better. Same thing for values below 1.0.

Thank you for your comment. We revised the sentence as suggested, changing the terms used as follows: “Values above 1.0 determine a greater change of Perme score in a shorter period of time and should be interpreted as a robust increase in the mobility status during patients’ ICU stay. On the other hand, values bellow 1.0 determine a small change of Perme score during patients’ ICU stay”.

What PMI has been seen in other ICU populations? Even if this metric is new, you can calculate it from the data provided in other studies.

Thank you for your comment. We did not calculate the Perme mobility index (PMI) in other ICU populations. It is true that we can calculate it from data provided from other studies; however, we will need to have access to the database in order to calculate the index accurately. As mentioned before, this is the first study proposing the Perme mobility index concept. Thus, we will need future studies to extrapolate this new concept for other populations and not be limited to COVID patients.

“ICU-AW” is inconsistently written.

Thank you for your comment. We revised all abbreviations for “ICU-acquired weakness” utilized in the manuscript adopting the abbreviation of “ICU-AW”. We identified a total of 3 (three) abbreviations inconsistently written and we corrected them.

What is the significance of the overall results with respect to the COVID-19 pandemic?

Thank you for your comment.

As suggested, the overall results with respect to the COVID-19 pandemic are included in the discussion section as follows: “Based on the study’s result, it was observed that COVID-19 pandemic not only causes respiratory impairment, but also affects patient’s mobility level. Although patients improve their mobility level, they do not achieve the highest Perme Score at ICU discharge, which demonstrates they still need post ICU rehabilitation. Another important aspect regarding the results in our study involves the associated factors to mobility level, such as age, comorbidities, and use of renal replacement therapy. This information may help physiotherapists to identify patients at higher risk of no improvement in mobility level and to include mobilization therapies earlier during ICU stay in this specific population, as therapy for COVID-19 patients should not only include respiratory therapy but also mobility therapy as early as possible. During this pandemic, there has been an increase in ICU demand, which may preclude the practice of adequate mobility therapy as in usual care. Therefore, it is interesting to stratify patients with higher risk of no improvement of mobility level during ICU stay.”

Will different variants have different effects on the change in mobility after ICU stay?

Thank you for your comment.

Unfortunately, we do not have specific data on different variants to evaluate its effect on the change in mobility after ICU stay. Future studies are still needed to further evaluate this matter.

Does this affect recommendations for in-ICU therapy or post-ICU therapy? See the general comments as well.

Thank you for your comment.

As described in previous comments, this study elucidates the importance of including mobility therapy associated with respiratory therapy in COVID-19 patients as early as possible as they do not achieve their higher mobility level at ICU discharge, which demonstrates the need for continuous rehabilitation, even after ICU discharge.

Limitations

One limitation is the limited time-frame of the study. It would have added more relevance if the patients had been followed post-ICU admission and post-discharge. You mention this in the last paragraph on page 14 but it should be mentioned as a limitation.

Thank you for your comment. We revised the content and rewrote the paragraph adding two limitations identified by the reviewers: 1) the timeframe of the study and 2) the fact the study was conducted only for COVID-19 patients.

The paragraph has been rewritten as follows: “Our study has limitations. First, the timeframe: it was limited to ICU stay only. The study might have had additional relevance if the patients had been followed post-ICU admission and post-discharge. Second, the focus on a specific population: this study focused on evaluating the mobility level of COVID-19 patients. However, in order to validate this new concept of the PMI, other populations must be studied. Information regarding the impact of mobility level during hospital stay as well as long-term follow-up should be investigated in future studies”.

Reviewer #3:

Dear authors

Thank you for this interesting study. I do have several comments.

Dear Reviewer #3,

Thank you for your precious time spent reviewing this manuscript. Below, you will find the answers point-by-point for all your comments and suggestions.

Ethics Statement: you write N/A but the study involves data of human participants. Then in the paper (page 4) you write that the consensus was waved so I guess you did get ethical approval? Please clarify. I do not know Brazilian regulations but in many countries, consensus would be required.

Thank you for your comment. We have added additional information on the sentence of the manuscript as follows: “The study was approved by the Institutional Review Board (IRB) – of Hospital Israelita Albert Einstein’s ethics committee under number CAAE: 30797520.6.0000.0071, and informed consent was waived”.

In the introduction

I would appreciate more info on comparable scales or indexes and the rationale why to develop the PMI.

Thank you for your comment. We have included more info on comparable scales or indexes and the rationale behind why to develop the PMI in the introduction as suggested: “Several measuring instruments have been adapted to assess the physical function of patients admitted to the ICU [13]. Currently a total of six measures have been specifically developed for ICU settings [13]: Chelsea Critical Care Physical Assessment Tool (CPAx) [14], Physical Function Intensive Care Unit Test Scored (PFIT-s) [15], the Perme Intensive Care Unit Mobility Score (Perme Score) [16], Intensive Care Unit Mobility Scale (IMS) [17], Intensive Care Unit Optimal Mobility Score (SOMS) [18], and the Functional Status Score for the Intensive Care Unit (FSS-ICU) [19]. All these assessment measures evaluate patients’ physical function; however, so far the Perme Score has been the only instrument to assess the barriers to mobilization. The study published by Wilches Luna et al. (2021) [20] established the minimal detectable change (MDC) of 1.36 points for the Perme Score, showing evidence of being sensitive to changes on patients’ mobility level [20]. Previous studies have calculated the minimal detectable change of some physical function instruments using the distribution‐based method; the MDC was calculated for PFIT‐s of 1.5 points [21], the CPAx established the value of 6.04 points [22], and the FSS‐ICU calculated the MDC of 2.0–5.0 points [23]. This determines the clinical significance of the results that are important to clinicians and researchers [20]. However, these findings do not consider the variation of the score during ICU length of stay (LOS) specifically; they only consider how many points of variation are needed to detect changes in the patients’ mobility status. Therefore, this study will evaluate the change of mobility level also considering the ICU LOS, thus testing a new concept entitled “Perme Mobility Index”. 

Also, because in case a scale already exists you need to compare. If it lacks then you have motive to develop yours.

Thank you for your comment.

On page 5 under "Mobility status assessment"

You mention that you included patients "with referral to physical therapy". Please expand: what are the indications or criteria to refer patients? Because this may create a bias in your sample.

Thank you for your comment. To avoid misinterpretation, the authors have decided to rewrite the sentence for readers’ better understanding as follows: “All consecutive patients admitted to the ICU with a confirmed diagnosis of COVID-19, who were assessed by a physical therapist, had their mobility status evaluated by the Perme Intensive Care Unit Mobility Score (Perme Score) (…)”

It is important to note that all the patients admitted to ICU with coronavirus (COVID-19) infection were assessed by a physical therapist due to the need for respiratory and motor evaluation and treatment. In Brazil, we do not have “respiratory therapist”, like in other countries in the world. So, the physical therapist is the professional responsible for managing patient’s respiratory conditions as well as for providing rehabilitation during ICU and hospital length of stay.

Information on the PT and/or larger Rehabilitation treatment of the patients is totally lacking. This would clearly influence the mobility of the patients and its evolution.

Thank you for your comment.

We have included information on the PT treatment of the patients as suggested: “All COVID-19 patients admitted to the ICU are assessed by the physiotherapy team for an initial evaluation. The present institution has an early mobility protocol and patients are seen every day by a physical therapist. The Perme Score is part of the daily mobility status evaluation in the early mobility protocol.”

Also, specific aspects regarding COVID-19 such as isolation, PPE use etc. are very linked and you provide no information at all. This may be very different from other populations with infectious disease. Also, the exercise tolerance of COVID-19 patients seems different from other ICU patients. Please mention that with some references.

Thank you for your comment.

We have included the specific aspects regarding COVID-19 as suggested in the Mobility Assessment Section: “Due to the need of isolation in COVID-19 patients, therapies were performed only around the ICU beds. Therefore, all ICU beds are individually isolated, with enough space to perform out of bed exercises (around 82 square feet) while maintaining the isolation during therapy.”

Statistical analysis

Here I have main concerns. As the Perme score is an ordinal scale it probably needs RASCH analysis. Given the data re not continuous you are not sure that one degree of improvement may value another and that you would be comparing the same difference.

Thank you for your comment.

All analyses of the Perme Score were based on two categories: “improved” and “not improved” rather than being analyzed as quantitative variable. Regarding the RASH analysis, we did not analyze Perme as continuous data. We always considered it categorized as 'improvement' vs. 'it doesn't get better. All analyses were based on two categories of Perme created according to what is recommended in the literature. The Perme Score was not used as a continuous variable as suggested.

Also, we need information on the clinically significant difference to decide when a patient is considered “Improved” or “not improved”.

Thank you for your comment. The present study defined two groups: 1) “Improved” – patients that presented PMI > 0, and 2) “Not improved” – patients that presented PMI ≤ 0. Unfortunately, one major limitation of the present study is that it was not be able to establish a clinically significant difference of this index. Future studies by our research team will focus on testing this clinimetric property.

How did you deal with patients who died? Did you have the scores at Death? are they considered as score at discharge?

Thank you for your comment. Patients who died were also included in the study; we have scores at death and yes, they were considered score at discharge.

In the Results section, page 9 first paragraph

I do not understand. You mean from 7 to 20?

Thank you for your comment. We apologize for the lack of clarity in the sentence. The context was given to present the difference of the mobility level assessed by the Perme Score at two distinct moments: 1) ICU admission and 2) ICU discharge. The values presented refer to the total points of the Perme score [median (IQR)] at the two moments.

After the revision, the sentence has been rewritten as follows: “The median (IQR) Perme Score total points in the pooled population study increased from ICU admission [7.0 (0.0–16) points] to ICU discharge [20.0 (7–28) points], respectively, P = < 0.001 (Table 1 and Fig 1)”.

Factors associated with mobility level: Data on therapy, Rehab treatment, PT etc are lacking.

Thank you for your suggestion. We agree that these factors may have an association with mobility level. However, we do not have these data to include in a multivariate analysis. If you find it necessary, we can include a sentence in the “Limitations” section of the manuscript.

Discussion

You need to compare with possibly existing other studies or other instruments on the same topic.

Thank you for your comment. We have included this topic with other instruments as suggested: “It is important to note that changes measured by using instruments over time can be confused with the clinimetric property entitled responsiveness – defined as the ability of an instrument to detect change over time [23]. There is also a minimum level of change that can be identified using an instrument called the minimal detectable change (MDC). It can be defined as the smallest alteration that the instrument is capable of detecting [24]. The study published by Wilches Luna et al. (2021) established the MDC of 1.36 points for the Perme Score, showing evidence of being sensitive to changes on patients’ mobility level [25]. This determines the clinical significance of the results important to clinicians and researchers [25].

The main difference between the PMI and MCD is that the index analyzes the variation of Perme Score at a specific moment in time. The MDC determines how many points of variation are needed to detect change in the patient mobility status.”

I do not understand your considerations on LOS page 12 in the second paragraph.

Thank you for the opportunity to clarify the aspects regarding the concept of the “Perme mobility index (PMI)”. The paragraph discusses particularities of the index. The index is calculated dividing the difference between the total Perme Score at ICU discharge minus the total Perme Score at ICU admission (�Perme Score) by the ICU length of stay. It is important to note that the PMI value can be either positive or negative.

[PMI = �Perme Score (Perme Score ICU DISCHARGE – Perme Score ICU ADMISSION) / ICU LOS]

Below we present three different examples with the same �Perme score of 14 points. We will adopt three different ICU length of stay for each example as follows: Example 1) ICU LOS = 14 days; Example 2) ICU LOS = 2 days, and Example 3) ICU LOS = 21 days, respectively.

• Calculating the �Perme Score:

Perme Score ICU DISCHARGE = 14 points

Perme Score ICU ADMISSION = 0 points

Perme Score = Perme Score ICU DISCHARGE – Perme Score ICU ADMISSION)

Perme Score = 14 – 0

Perme Score = 14

By calculating the index value, for each example we will find the following values: Example 1 >>> PMI = 1.0; Example 2 >>> PMI = +7.0; and Example 3 >>> PMI = +0.67.

• Example 1:

ICU LOS = 14 days

[PMI = �Perme Score (Perme Score ICU DISCHARGE – Perme Score ICU ADMISSION) / ICU LOS].

PMI = (14 - 0) / 14

PMI = (14) / 14

PMI = +1.0

• Example 2:

ICU LOS = 2 days

[PMI = �Perme Score (Perme Score ICU DISCHARGE – Perme Score ICU ADMISSION) / ICU LOS].

PMI = 14 / 2

PMI = +7.0

• Example 3:

ICU LOS = 21 days

[PMI = �Perme Score (Perme Score ICU DISCHARGE – Perme Score ICU ADMISSION) / ICU LOS].

PMI = 14 / 21

PMI = +0.67

Note that we adopted the same �Perme Score, yet in each case the ICU LOS was different. The PMI calculation in example “#1” presented the value equal “+1.0”, which means that the change in the Perme score from ICU admission to discharge is the same as the ICU length of stay. The positive value “+” indicates that the patient improved the mobility level at discharge compared to the mobility level at ICU admission. However, if the index value was negative “-”, it indicated that the patient decreased the mobility level from ICU admission to ICU discharge.

In example “#2”, the calculation of PMI presented the value of “+7.0”. This value can be interpreted as a robust change in the Perme score in patient in a short period of time. Although the �Perme Score is the same as in example “#1”, in this hypothetical case the variation occurred in just 2 days, which shows the influence of the ICU LOS on the calculation of the index. The PMI calculation in example “#3” presented the value of “+0.67”. We utilized the same �Perme Score of 14 and the ICU LOS of 21 days. The change of the Perme Score was the same as in examples “#1” and “#2”, but the ICU LOS influenced the value of PMI below “1.0”, indicating that the variation of the Perme score (�Perme Score) was smaller than the ICU LOS.

Thus, we recommend using the dimensionless reference value of “1.0” as a starting point for PMI analysis. For example, the PMI value of “1.0” should be carefully interpreted, considering that the patient can present a �Perme Score of 32, indicating the maximum change of Perme score, starting from the initial value of “0” and culminating in the maximum value of “32”. This is a very good result for a total ICU LOS of 32 days. This means that the patient improved the mobility level considerably but at the price of a long hospital length of stay.

We hope we have clarified the topic regarding the PMI and the length of stay presented in the second paragraph of page 12.

Limitations: the fact that this study considers only COVID patients which means a very specific context; would need to be validated also in other populations.

Thank you for your comment. We agree that the findings of this study are limited to COVID-19 patients admitted to the ICU. A new paragraph has been added in the manuscript, identifying the limitations of this study, as follows: “Our study has limitations. First, the timeframe: it was limited to ICU stay only. The study might have had additional relevance if the patients had been followed post-ICU admission and post-discharge. Second, the focus on a specific population: this study focused on evaluating the mobility level of COVID-19 patients. However, in order to validate this new concept of the PMI, other populations must be studied. Information regarding the impact of mobility level during hospital stay as well as long-term follow-up should be investigated in future studies”.

 Reviewer #4:

This was a retrospective observational study of 136 patients admitted to an ICU in single-center in Brazil with COVID. The authors found an association between improved mobility scores and better outcomes. Predictors of better mobility scores were identified. The included patients were divided into two groups based on a novel scoring introduced in this study called the PMI. The results were presented by PMI group.

Dear Reviewer #4,

Thank you for your comments and suggestions on this manuscript. We appreciate your hard work on reviewing this manuscript. See below the answers for your comments 

An exclusion criteria is patients without mobility status report at ICU admission and/or discharge. There is a potential for significant selection bias by excluding these patients. For example, patients who died during ICU stay may not have a mobility status at “ICU discharge”.

Thank you for your comment.

The patients excluded from the study were patients that did not have mobility status recorded on their electronic chart; but that was not because they died, as patients who died were also included in the study and their score was considered the score on the day they died.

To clarify a few issues regarding patients who died during ICU stay, it is important to emphasize that the ICU mortality rate of included patients was 16.2% (n = 22/136). All the deaths included in the group of “not improved” patients had a mobility level “0 – zero” assessed for these patients at the moment of “ICU discharge”. In order to avoid misinterpretation of the results and to provide additional information of excluded patients, we added a new table at “Supporting information” section, entitled: “Supporting Information Table 2 – Clinical outcomes in patients with or without missing in Perme Score”.

It seems that only patients who were referred to physical therapy during ICU stay had their mobility levels assessed using the Perme score and were included in this study. Patients who were not referred to physical therapy were excluded. This introduces a potential source of selection bias that must be addressed.

Thank you for your comment. After a careful revision of the “Materials and methods” section, we realized that the inclusion criteria are not clear enough, inducing a potential source of selection bias, as identified by the reviewer. We thank you for the opportunity to clarify the inclusion criteria of the study. See below the paragraph of inclusion and exclusion criteria adopted:

“The clinical records of the first 200 consecutive patients admitted to the ICU with diagnosis of COVID-19 confirmed by reverse transcription–polymerase chain reaction (RT-PCR) for SARS-CoV-2 were considered eligible for the study. Key eligibility criteria included the following: 1) admission to the ICU and 2) equal or older than 18 years. Study exclusion criteria consisted of patients who did not have a mobility status report on their electronic medical record at ICU admission and/or discharge”.

We have also rewritten the following sentence to clarify the data collection of mobility status using the Perme Score: “All consecutive patients admitted to the ICU with a confirmed diagnosis of COVID-19, who were assessed by a physical therapist, had their mobility status evaluated by the Perme Intensive Care Unit Mobility Score (Perme Score) (…)”

There were 64 (32%) patients excluded from the study due to missing data. Comparing the clinical outcomes in these patients (without mobility data) to those included in the study (with mobility data) would be helpful at addressing the above points. Was the mortality rate or length of stay similar?

Thank you for your comment and suggestion to run the analysis comparing the clinical outcomes of the two groups. We decided to add two new tables at “Supporting information” section, entitled:

• “Supporting Information Table 2 – Baseline characteristics of patients with or without missing in Perme Score”.

• “Supporting Information Table 3 – Clinical outcomes in patients with or without missing in Perme Score”.

The median (IQR) ICU length of stay in an overall analysis was lower in patients allocated in the “Missing group” compared with “No missing group”, which was 3.0 (2.0–3.0) days versus 12.0 (7.0–23.2); P < 0.001, respectively. The same result is observed in survivors’ analysis. The median (IQR) ICU length of stay was lower in patients allocated in the “Missing group” compared with “No missing group”, which was 3.0 (2.0–4.0) days versus 11.0 (6.2–21.0); P < 0.001, respectively.

Analyzing the hospital length of stay, the median (IQR) in an overall analysis was lower in patients allocated in the “Missing group” compared with “No missing group”, which was 8.5 (5.0–12.2) days versus 19.5 (12.2–35.0); P < 0.001, respectively. The same result is observed in survivors’ analysis. The median (IQR) hospital length of stay were lower in patients allocated in the “Missing group” compared with “No missing group”, which was 8.0 (5.2–12.0) days versus 19.5 (12.0–35.2); P < 0.001, respectively.

There was no difference in ICU mortality rate in an overall analysis in the “Missing group” compared with “No missing group”, which was 7.0 (10.9) versus 22 (16.2); P = 0.393, respectively. The same result is observed in survivors’ analysis. The n (%) ICU mortality rate did not present any difference when comparing the “Missing group” with “No missing group”, 10.0 (15.6) versus 22.0 (16.2); P = 0.999, respectively. No difference was observed in hospital mortality rate in survivors’ analysis. The n (%) hospital mortality rate of “Missing group” compared with “No missing group” was 10.0 (15.6) versus 22.0 (16.2); P = 0.999, respectively.

A shorter ICU and hospital length of stay was observed in the “missing in Perme” group. In the “Missing in Perme” group, patients had a lower SAPS III and SOFA score compared with the “No Missing” group as well as the need for noninvasive and invasive ventilation. Additional information is available at “Supporting Information Table 2”. These findings support the lower ICU and hospital LOS observed.

How about other key clinical datapoints?

Thank you for your comment.

Other key clinical datapoints were included in the Discussion section as follows: “Based on the study’s result, it was observed that COVID-19 pandemic not only causes respiratory impairment, but it also affects patient’s mobility level. Although patients improve their mobility level, they do not achieve the highest Perme Score at ICU discharge, demonstrating that these patients still need post ICU rehabilitation. Another important aspect regarding the results in our study involves the factors associated with mobility status, such as age, comorbidities, and use of renal replacement therapy. This information may help physiotherapists to identify patients at higher risk of no improvement in mobility level and to include mobilization therapies earlier during ICU stay in this specific population, as therapy for COVID-19 patients should not only include respiratory therapy but also mobility therapy as early as possible. During this pandemic, there has been an increase in ICU demand, which may not allow for an adequate mobility therapy as in usual care. Therefore, it is interesting to stratify patients with higher risk of no improvement of mobility level during ICU stay.”

The methods state that a “convenience sample” was used. If the study was done retrospectively with ethics approval, why were consecutive patients not included?

Thank you for your comment. The sample size was based on the first 200 consecutive cases of patients admitted to the ICU with confirmed diagnosis of COVID-19 at Hospital Israelita Albert Einstein. We realized that we did not mention the “first 200 consecutive cases” on the first version of the manuscript. Thus, we have rewritte the sentence as follows: “A convenience sample of the first 200 consecutive patients admitted to the ICU with a confirmed diagnosis of COVID-19 was considered for this analysis”.

Page 10: This sentence seems to be missing a final word: “No multicollinearity was observed in the final model, data presented in (S1 Table).”

Thank you for your comment. Yes. We corrected the sentence adding the missing word, as follows: “No multicollinearity was observed in the final data model presented in Supporting Information 1 Table (S1 Table)”.

Page 14: Should be “purpose” in the sentence: “The change of mobility status over time is the main propose of the PMI.

Thank you for your comment. Yes. We definitely overlooked the typo. We have corrected the sentence as suggested: “The change of mobility status over time is the main purpose of the PMI”.

---

## [Editor Report · Decision Letter 1]

1 Apr 2021

The Perme Mobility Index: a new concept to assess mobility level in patients with coronavirus (COVID-19) infection.

PONE-D-21-01443R1

Dear Dr. Nawa,

We’re pleased to inform you that your manuscript has been judged scientifically suitable for publication and will be formally accepted for publication once it meets all outstanding technical requirements.

Kind regards,

Aleksandar R. Zivkovic

Academic Editor

PLOS ONE

---

## [Editor Report · Acceptance letter]

12 Apr 2021

PONE-D-21-01443R1 

The Perme Mobility Index: a new concept to assess mobility level in patients with coronavirus (COVID-19) infection 

Dear Dr. Nawa:

I'm pleased to inform you that your manuscript has been deemed suitable for publication in PLOS ONE. Congratulations! Your manuscript is now with our production department. 

Kind regards, 

on behalf of

Dr. Aleksandar R. Zivkovic 

Academic Editor

PLOS ONE